# The Antiproliferative Activity of High-Dose Somatostatin Analogs in Gastro-Entero-Pancreatic Neuroendocrine Tumors: A Systematic Review and Meta-Analysis

**DOI:** 10.3390/jcm11206127

**Published:** 2022-10-18

**Authors:** Francesco Panzuto, Claudio Ricci, Maria Rinzivillo, Ludovica Magi, Matteo Marasco, Giuseppe Lamberti, Riccardo Casadei, Davide Campana

**Affiliations:** 1ENETS Center of Excellence, Disease Unit, Sant’Andrea University Hospital, 00189 Rome, Italy; 2Department of Medical-Surgical Sciences and Translational Medicine, Sapienza University of Rome, 00189 Rome, Italy; 3Division of Pancreatic Surgery, IRCCS Azienda Ospedaliero Universitaria di Bologna, 40138 Bologna, Italy; 4Alma Mater Studiorum, Department of Internal Medicine and Surgery (DIMEC), S. Orsola-Malpighi Hospital, University of Bologna, 40138 Bologna, Italy; 5Alma Mater Studiorum, Department of Internal Specialized, Diagnostic and Experimental Medicine (DIMES), S. Orsola-Malpighi Hospital, University of Bologna, 40138 Bologna, Italy; 6Division of Oncology, IRCCS Azienda Ospedaliero Universitaria di Bologna, 40138 Bologna, Italy

**Keywords:** neuroendocrine neoplasms, high dose of somatostatin analogs, meta-analysis

## Abstract

Background: The antiproliferative activity of a high dose of somatostatin analogs (HD-SSA) in treating gastro-entero-pancreatic neuroendocrine neoplasms (GEP-NENs) remains under debate. Methods: A systematic review and proportion meta-analysis were made. The primary endpoint was the efficacy measured as incidence density ratio (IDR) at one year. The secondary endpoints were the disease control rate (DCR) and severe adverse events (SAEs). The heterogeneity (I2), when high (>50%), was interpreted by performing a univariate metaregression analysis, analyzing as covariates: type and design of the study, location (Europe or USA), sample size, grading according to 2017 WHO, the metastatic disease rate, previous therapy including surgery, and quality of the study. Results: A total of 11 studies with 783 patients were included. The IDR was 62 new progressions of 100 patients treated with HD-SSA every one year. The heterogeneity was high. The study’s year, type and design, primary tumor, grading, previous treatments, and quality of the studies did not influence the IDR. The IDR was significantly higher in USA centers and studies with more than 50 patients. The IDR was lower when a high rate of metastatic patients was present in the studies. The DCR was 45%. The heterogeneity was high. The DCR was lower in USA studies and in prospective trials. Conclusion: Given the limited efficacy of HD-SSA in preventing the disease progression in unresectable GEP-NENs after failure of standard dose SSA, the use of this therapeutic approach is advisable in selected cases when other antiproliferative treatments are not feasible.

## 1. Introduction

According to the WHO classifications, gastro-entero-pancreatic neuroendocrine neoplasms (GEP NENs) are classified based on tumor morphology (well-differentiated neuroendocrine tumors—NET—vs. poorly differentiated neuroendocrine carcinomas—NEC) and grading, which is usually assessed by Ki67 proliferative index (G1 = Ki67 < 3%, G2 = Ki67 3–20%, G3 = Ki67 > 20%) [1,2]. Disease aggressiveness is affected by several factors, including primary tumor site, grading, stage, tumor burden, somatostatin receptors expression, and metabolic activity (assessed by FDG-PET), which are used for evaluating patient’s prognosis and for planning the optimal medical treatment when curative surgery is not feasible due to advanced disease [3,4,5]. Treatment options for patients with NEN are continuously expanding and include long-acting somatostatin analogs (SSAs), peptide receptor radionuclide therapy (PRRT), tyrosine kinase inhibitors, mTOR inhibitors, and systemic chemotherapy [6].

Long-acting SSAs octreotide and lanreotide are widely considered effective and well-tolerated first-line treatment for G1-G2 GEP NETs expressing sstr, following the results of the phase-3 randomized controlled trials PROMID and CLARINET [7,8]. There are five types of sstr (sstr1–5) whose activation by native somatostatin or SSAs results in antiproliferative effects on tumor cells via direct and indirect mechanisms. Activation of sstr on tumor cells leads to cell cycle arrest and apoptosis through regulation of MAP kinase and phosphotyrosine phosphatase activity, while indirect mechanisms involve the angiogenesis and growth factor secretion inhibition.

SSAs compare favorably with the other approaches available for the treatment of NETs: indeed, SSAs have an excellent toxicity profile and are well-tolerated by patients (with mild gastrointestinal symptoms which are usually transient), have a convenient administration schedule, can control symptoms in patients with hormone-producing tumors, and have shown to have antiproliferative effect. Standard SSA dose is one single injection every 4 weeks, at the standard doses of 30 mg and 120 mg for octreotide and lanreotide, respectively. While native somatostatin binds all sstr types but type 5 (sstr5) at nanomolar concentrations, both SSAs selectively bind with high affinity type 2 sstr (sstr2), which is preferentially expressed on NETs, and with slightly lower affinity sstr5. Their potential increased antiproliferative activity, when used at higher doses in patients not responding to standard dose SSA, has been investigated over the last two decades by several retrospectives or small prospective studies, which report promising results in terms of disease control rates (widely ranging from 30% to 100%) and median progression-free survival (PFS) value (up to 32 months) [9,10,11]. However, a less favorable outcome, with a median PFS of 8.4 months, was observed with high-dose octreotide (60 mg/4 weeks) in the control group arm of the NETTER-1 study, which was designed to investigate the efficacy of 177Lu-DOTATATE vs. high dose octreotide in midgut NETs [12]. Similarly, in the recently reported phase-2 CLARINET FORTE trial which, patients receiving doubled dose lanreotide (120 mg/2 weeks) after progressing on the standard dose (120 mg/4 weeks) had a median PFS of 8.3 months and 5.8 months in pancreatic and midgut NETs, respectively [13].

Given the heterogeneity of available data on the antiproliferative activity of high-dose (HD) SSAs, we undertook this systematic review and meta-analysis to assess the current literature regarding the efficacy of increasing octreotide or lanreotide dose in patients with progressive GEP NET after standard dose treatment.

## 2. Materials and Methods

The manuscript was structured following the Preferred Reporting Items for Systematic Reviews and Meta-analysis (PRISMA) statement [14].

### 2.1. Eligibility Criteria, Information Sources, and Search

All studies fulfilling the following PICOS criteria [15] were considered eligible for the present study:Population (P): patients with unresectable GEP-NENs;Interventions (I): HD-SSA;Comparator (C): none;Outcome (O): Progression-free survival (PFS), disease control rate (DCR), and Severe Adverse Events (AEs);Studies: prospective and retrospective studies.

Studies were included when Kaplan–Meier of PFS was reported. Review articles without original data and case reports were excluded. A systematic review of the literature was conducted following the recommendations for systematic reviews in surgery provided by Goossen et al. [16]. The PubMed databases were searched for eligible articles in the English language without publication date or publication type restriction. The last search was carried out on 27 October 2021. The search was conducted using medical subject headings (MeSH) combined with the following non-MeSH words. The string search used in MEDLINE/PubMed was: (neuroendocrine tumors [MeSH Terms]) AND (octreotide or lanreotide) OR (somatostatin [MeSH Terms]).

### 2.2. Study Selection and Data Collection Process

The identified records were screened for title and abstract independently by two investigators (M.R. and G.L.). If the paper was considered eligible, the full-text text was evaluated. Data were extracted from the selected articles using a prefixed electronic form. Extracted data were then compared, and any discrepancies were solved through discussion. Any disagreement regarding inclusion criteria was solved through discussion or consulting the last author (D.C.). The PRISMA flow diagram was reported in Figure 1.

### 2.3. Data Items, Risk of Bias in Individual Studies, Summary Measures, and Synthesis of Results

The following data were extracted to describe the included studies: year of publication, first author, study type and design, study period, institution and country, study period, number of participants, type, and schedule of SSA. Tumor origin, grading according to the 2017 World Health Organization (WHO) classification [1], metastases, previous treatment with the standard dose of SSA or chemotherapy or surgery of primary tumors were also extracted to evaluate the influence on the outcomes. The quality of studies was assessed with the Risk Of Bias In Non-randomized Studies—of Interventions (ROBINS-I) tool [17]. Incidence density rate (IDR) was used to standardize PFS measurement among the different studies. IDR represents the number of events for at-risk patients per year and makes comparable studies with different observation times. These measures can be assimilated to the hazard rate every year for patients exposed [18]. Thus, the ratio obtained from the IDR incidence density rates can be assimilated to the HR only for the exponential model (constant hazard functions) and the absence of significant differences in the average follow-up duration between the sub-groups [18]. To obtain the crude number of events and observation period from Kaplan–Meier curves, we used dedicated software (GetData Graphical Digitizer@). The results were reported as a pooling proportion (effect size) and a 95% CI using a random effect model. The meta-analysis was carried out in line with recommendations from the Cochrane Collaboration and Meta-analysis of Observational Studies in Epidemiology guidelines [19,20] and the Mantel–Haenszel random-effects model was used to calculate the effect size [21].

### 2.4. Risk of Bias across Studies and Additional Analyses

The risk of bias across included studies was measured using the *I*^2^, which describes the variability in point estimates due to heterogeneity rather than sampling error [22]. When *I*^2^ was <50%, the risk of “between-study heterogeneity” was judged as low-moderate; if *I*^2^ was ≥50%, the risk of “between-study heterogeneity” was considered high. The meta-regression analysis was performed when heterogeneity was high. The meta-regression was performed using the maximum residual likelihood (REML) approach [23]. The values obtained from metaregression represent the HRs for PFS and the ORs for the DCR and severe AEs, obtained comparing the subgroups. R^2^ indicates the heterogeneity explained by the covariate. A *p* value < 0.05 was considered statistically relevant.

The statistical analysis was carried out using dedicated packages for STATA version 14^®^ (StataCorp, College Station, TX, USA).

## 3. Results

### 3.1. Study Selection

Article selection process is shown in Figure 1. A total of 19,283 articles were screened, but only 29 studies were evaluated in full-text form. Of these, 18 were excluded because they did not meet inclusion criteria. Finally, only 11 papers [10,11,12,13,24,25,26,27,28,29,30] were considered suitable for the meta-analysis.

### 3.2. Study Characteristics and Risk of Bias within Studies

All the papers were published between 1994 and 2021. Eleven studies involving 783 patients were included. There were eight retrospective and three prospective cohorts. The majority of the studies (63.6%) were multicentric and conducted exclusively in European countries. The median sample size of the studies was 54 (range 12–140). The different schedules used are reported in Table 1. The majority of studies (81.8%) have a moderate risk of bias. The other potentially relevant confounding factors are reported in Appendix A.

The meta-analytic results are reported in Table 2 and Figure 2, Figure 3 and Figure 4. The proportion of patients who experienced a disease progression was 62% (53 to 70, 95% CI) per 100 subjects treated every year. Pooled DCR and severe Aes rates were 45% (24 to 64, 95% CI) and 9% (3% to 14%, 95% CI), respectively. All results are affected by high heterogeneity: 96.4%, 98.1%, and 88%, for PFS, DCR, and severe Aes.

At univariate meta-regression analysis, PFS was significantly influenced by three factors (Table 3). The risk of progression was significantly higher in the studies coordinated by USA centers (HR 1.23; 1.03 to 1.45; *p* = 0.021) and when more than 54 patients were enrolled (HR 1.31; 1.04 to 1.64; *p* = 0.023, Figure 5). 

The risk of recurrence was lower in studies with a high rate of metastatic patients (HR 0.35; 0.14 to 0.87; 95% CI). The DCR rate was lower in USA studies (OR 0.76; 0.59 to 0.98; *p* = 0.040) and in prospective trials (0.77; 0.67 to 0.89; *p* < 0.003). No factor explained the heterogeneity of severe Aes.

## 4. Discussion

SSAs are the mainstay of treatment of well-differentiated GEP-NET since they showed antiproliferative effect [7,8] with a very tolerable safety profile [31,32]. Subsequent treatment lines at progression include PRRT, TKIs, or chemotherapy, all of which have a toxicity profile less favorable than SSAs, with PRRT preferred over other alternatives [33,34]. Non-conventional doses SSA (HD-SSA), achieved by either dose density or dose intensity increase, have been proposed and investigated as a potential treatment option in patients with GEP-NET whose disease progressed on standard dose SSA. Well-differentiated NETs are a heterogeneous group of tumors whose prognosis varies hugely based on baseline clinicopathological variables, including previous evidence of radiological progression, primary organ of origin, Ki67, and extra-hepatic involvement among others, as showed by the highly different median PFS observed in the CLARINET and PROMID studies of lanreotide autogel and octreotide, respectively [7,8], and in the studies investigating the role of HD-SSA (with median PFS ranging from 5 to 30 months). Our meta-analysis showed a relatively high proportion of patients who experience disease progression per year while on HD-SSA, with a discrete rate of DCR as best response and a low incidence of severe adverse events. However, the studies included in the present meta-analysis are highly heterogenous, as captured by an *I*^2^ of approximately 90% and above. To investigate this aspect, a metaregression was performed that showed that DCR is lower in prospective studies than in retrospective ones and in those carried out in the USA compared to those carried out in Europe, with the latter applying also to PFS. An explanation to these findings might be that studies with more rigorous tumor response assessment criteria are more likely to identify and report earlier progressive disease. On the other hand, PFS is shorter in studies with less metastatic patients likely because progressive disease might be more difficult to identify in patients with multiple metastasis, e.g., miliary liver involvement or type III pattern [35]. Furthermore, PFS is shorter for studies with greater sample size, possibly because it tracks with better-conducted, more rigorous studies, with stricter criteria for tumor response evaluation assessment and report of disease progression.

Reviewing the most significant studies, it was found that in the CLARINET FORTE phase II trial of lanreotide 120 mg every 14 days in patients with midgut (N = 79) or pancreatic NET (N = 79), a dramatic decrease in median PFS was observed in tumors with Ki67 >10% as compared to those with a lower proliferation index, in both the midgut (5.5 vs. 8.6 months, respectively) and the pancreatic cohort (2.8 vs. 8.0 months, respectively) [13]. Findings from the CLARINET FORTE trial are in line with the findings of our meta-analysis as it shows that HD-SSA is a feasible treatment option with acceptable PFS outcome only in a subset of patients with pancreatic NET progressing on standard dose SSA, namely those with ki67 ≤ 10% as per the post-hoc analysis of the trial. In a retrospective UK series of 105 patients with GEP-NET who each received either lanreotide autogel 120 mg or octreotide 30 mg every 3 weeks, median PFS was 25 months ad it was shorter in patients with PFS < 12 months to previous standard-dose SSA treatment, pancreatic primary, Ki-67 ≥ 5% and extrahepatic metastases [11]. However, in this study 58% of patients received HD-SSA because of symptoms progression and 11% because of elevation in serum biomarkers, which could have selected for more indolent disease on the radiological progression side and explain the long PFS observed. Nevertheless, in a large Italian multicenter retrospective study that included 140 patients with GEP-NET who received HD-SSA upon radiological progression to previous treatment, a median PFS of 31 months in the overall cohort was observed [10]. Furthermore, the median PFS was longer when HD-SSA was used as second-line treatment as compared to later lines of treatment, with a trend toward an association with previous standard dose SSA treatment duration, similarly to that observed in the UK series [11].

## 5. Conclusions

In conclusion, available literature and the results of our meta-analysis suggest that HD-SSA is not the preferred treatment choice in patients with GEP-NET who progressed on standard-dose SSA because of the short PFS and low DCR reported, especially when compared with other alternatives, such as PRRT [12,33]. This is markedly more evident in studies carried out in the USA, with prospective design, and in patients with metastatic disease. However, a subset of patients with advanced age, whose disease showed indolent behavior, long PFS on standard-dose SSA (>12 months), low Ki67/grading, and low or no extrahepatic metastatic burden, could benefit from HD-SSA treatment as a low-toxicity effective treatment that can preserve quality of life.

## Figures and Tables

**Figure 1 jcm-11-06127-f001:**
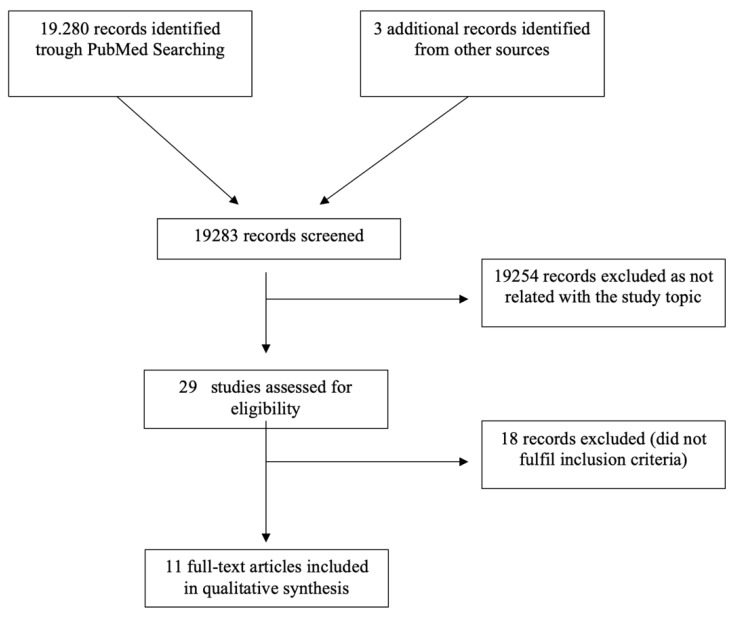
Results of search and reasons for exclusion of papers according to PRISMA statement.

**Figure 2 jcm-11-06127-f002:**
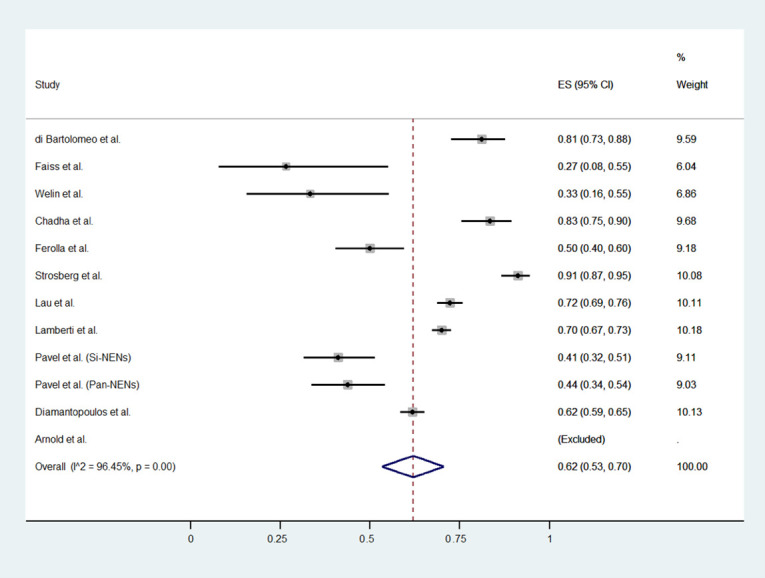
Forest plot for pooled incidence density rate of PFS in patients treated with HD-SSA [10,11,12,13,24,25,26,27,28,29,30].

**Figure 3 jcm-11-06127-f003:**
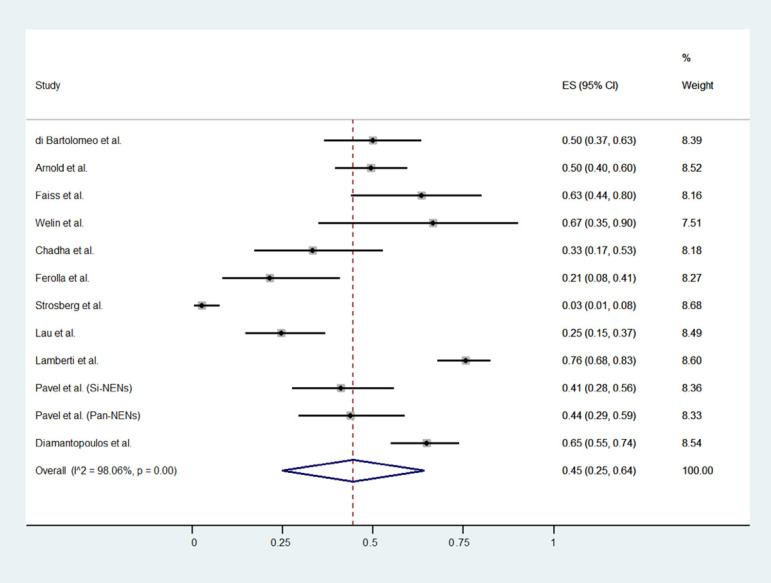
Forest plot for pooled incidence density rate of DCR in patients treated with HD-SSA [10,11,12,13,24,25,26,27,28,29,30].

**Figure 4 jcm-11-06127-f004:**
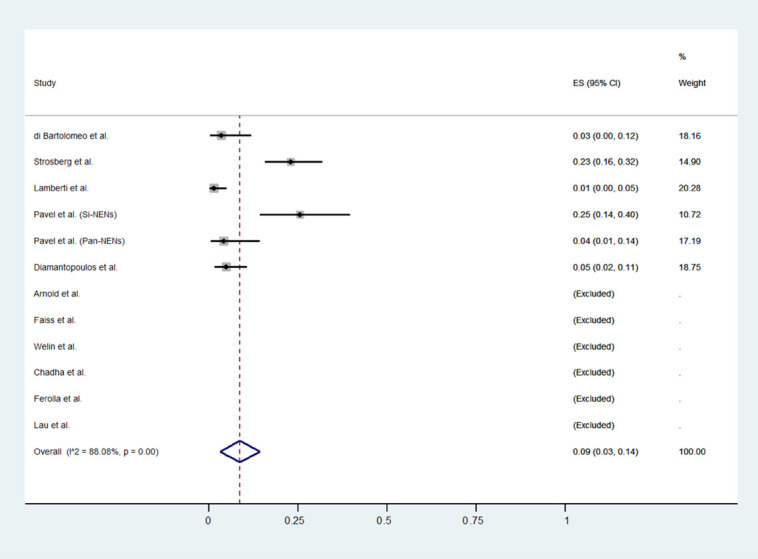
Forest plot for pooled incidence density rate of Aes in patients treated with HD-SSA [10,11,12,13,24,25,26,27,28,29,30].

**Figure 5 jcm-11-06127-f005:**
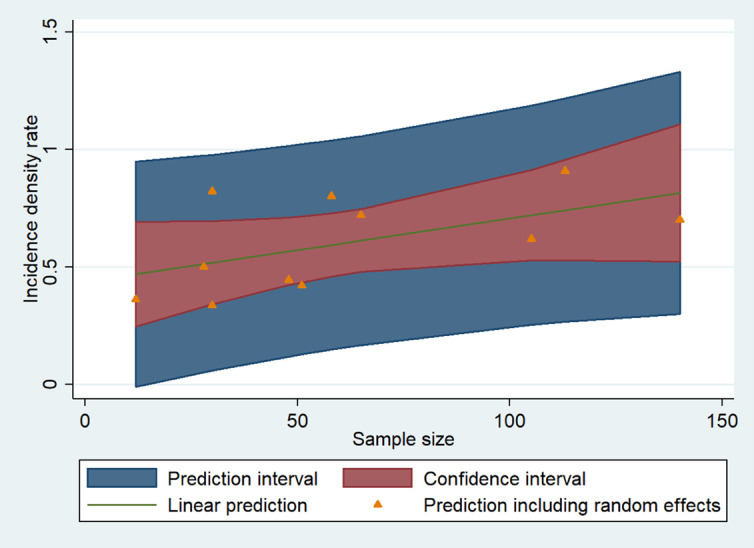
Results of metaregression analysis: the relationship between the incidence density rate and sample size of the studies.

**Table 1 jcm-11-06127-t001:** Characteristics of 11 included studies.

Year	Authors	Study Type	Study Design	Center(s) (Country)	Study Period	Patients Enrolled	Therapy	ROBINS-I
1994	Di Bartolomeo et al. [24].	Retrospective	Multicentric	13 Italian centers	1992–1994	58	1.5 mg daily; 3 mg daily	Moderate
1996	Arnold et al. [25].	Retrospective	Multicentric	49 German centers	1989–1991	103	1.5 mg daily	Moderate
1999	Faiss et al. [26].	Retrospective	Multicentric	3 German centers	Not reported	30	15 mg daily ^	Moderate
2004	Welin et al. [27].	Retrospective	Monocentric	Sweden	Not reported	12	160 mg every 2–4 week °	Moderate
2009	Chadha et al. [28].	Retrospective	Monocentric	USA	2002–2007	30	>30 mg every month ^§^	Moderate
2012	Ferolla et al. [29].	Prospective	Multicentric	Italy	2007–2008	28	30 mg every 3 week ^§^	Moderate
2017	Strosberg et al. [12].	Prospective	Multicentric	41 centers, 8 countries world-wide	2012–2016	113	60 mg every 4 week ^§^	Low
2018	Lau et al. [30].	Retrospective	Monocentric	Canada	2000–2013	65	>30 mg every month ^§^	Moderate
2019	Lamberti et al. [10].	Retrospective	Multicentric	Italy	2004–2017	140	180 mg every 4 week ^ or 60 mg every 4 week ^§^	Moderate
2021	Pavel et al. [13].	Prospective	Multicentric	25 European centres	2015–2019	51 + 48 ^#^	120 mg every 14 days ^	Low
2021	Diamantopoulos et al. [11].	Retrospective	Monocentric	UK	2003–2017	105	120 mg every 21 days ^	Moderate

Legend: ^ = Lanreotide; ° = Octeotride Pamoato: ^§^ = Octreotide Acetato; ^#^ = 51 midgut and 48 pancreatic endocrine neoaplasm (panNET); PFS = progression free-survival; DCR = Disease Control Rate; Severe AEs = Severe Adverse Events.

**Table 2 jcm-11-06127-t002:** Results of meta-proportion analysis.

Endpoints	Number of Studies	Effect Size (95% CI)	*p*-Value	Heterogeneity I^2^ (%)
PFS	11 ^§^	0.62 (0.53 to 0.70)	<0.001 *	96.4
DCR	11 ^§^	0.45 (0.24 to 0.64)	<0.001 *	98.1
Severe Aes	11 ^§^	0.09 (0.03 to 0.14)	<0.001 *	88

Legend: * = the referent for effect size is the zero value; when *p*-value is <inferior to 0.05, the event is statistically significant; PFS = progression-free survival; DCR = disease control rate; ^§^ = 11 studies counting 12 cohorts were included because Pavel et al. [13] reported pancreatic and small intestinal NENs results separately.

**Table 3 jcm-11-06127-t003:** Univariate meta-regression analysis for incidence density ratio, disease control rate, severe adverse events.

Covariates		PFS			DCR			Severe Aes		
	Number of Studies	HR (95 CI)	R^2^ (%)	*p*-Value	OR (95 CI)	R^2^ (%)	*p*-Value	OR (95 CI)	R^2^ (%)	*p*-Value
Publication year (before vs. after 2000)	11	1.07 (0.77 to 1.50)	0	0.570	0.88 (0.63 to 1.23)	0	0.404	1.08 (0.80 to 1.50)	0	0.570
Study type (retrospective vs. prospective)	11	1.10 (0.87 to 1.39)	5	0.350	0.77; 0.67 to 0.89	16	0.003	1.10 (0.87 to 1.40)	5	0.305
Study design (multicenter vs. uncenter)	11	1.06 (0.76 to 1.49)	0	0.655	0.97 (0.70 to 1.33)	0	0.825	1.06 (0.76 to 1.49)	0	0.655
Study coordinator center (Europe vs. USA)	11	1.23 (1.04 to 1.45)	41	0.023	0.77 (0.67 to 0.89)	60	0.003	1.08 (0.95 to 1.24)	53	0.157
Sample of size (≤ or >median value)	11	1.31 (1.04 to 1.63)	41	0.023	1.10 (0.74 to 1.35)	0	0.990	0.94 (0.72 to 1.24)	0	0.586
Rate of Pan-NENs (increasing)	11	0.91 (0.52 to 1.59)	0	0.109	1.01 (0.58 to 1.80)	0	0.944	0.87 (0.63 to 1.21)	4	0.310
Rate of Si-NENs (increasing)	11	1.05 (0.64 to 1.71)	0	0.839	0.93 (0.61 to 1.44)	0	0.839	1.18 (0.90 to 1.56)	45	0.143
Rate of CR-NENs (increasing)	11	0.26 (0.01 to 7.31))	11	0.372	7.01 (0.57 to 87.5)	25	0.110	0.33 (0.01 to 11.5)	0	0.397
G1 ^ (increasing rate)	6	1.37 (0.56 to 3.32)	0	0.401	0.80 (0.24 to 2.7)	0	0.668	1.45 (0.69 to 3.06)	32	0.212
Metastatic patients (increasing rate)	11	0.36 (0.14 to 0.87)	36	0.029	1.13 (0.34 to 3.73)	0	0.814	1.69 (0.77 to 3.70)	36	0.136
Previous SD of SSA (increasing rate)	11	1.29 (0.78 to 2.12)	3	0.987	1.01 (0.67 to 1.51)	0	0.989	1.42 (0.29 to 6.99)	0	0.570
Previous chemotherapy (increasing rate)	9	0.95 (0.06 to 15.2)	0	0.840	0.98 (0.19 to 4.99)	0	0.980	0.46 (0.01 to 51.73)	0	0.555
Surgery of primary tumors (increasing rate)	10	1.12 (0.43 to 2.86)	0	0.794	0.89 (0.45 to 1.74)	0	0.719	1.01 (0.46 to 2.17)	0	0.995
Risk of bias (low vs. moderate)	12	1.02 (0.72 to 1.44)	0	0.893	1.24 (0.92 to 1.68)	14	0.145	0.87 (0.73 to 1.05)	43	0.118

Legend: PFS = Progression-free survival; DCR = Disease control rate; ADE = adverse events; HR = Hazard Ratio; OR = Odds Ratio; R^2^ = % of heterogeneity explained by covariate; USA = United States of America; Pan-NENs = pancreatic neuroendocrine neoplasms; Si-NENs = small intestinal neuroendocrine neoplasms; CR-NENs = Colo-Rectal neoplasms; ^ = grade according to 2017 WHO classification; SD = Standard Dose; SSA = Somatostatin Analogs.

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
