# Peer review of "The Antiproliferative Activity of High-Dose Somatostatin Analogs in Gastro-Entero-Pancreatic Neuroendocrine Tumors: A Systematic Review and Meta-Analysis"

_jcm, 2022, doi:10.3390/jcm11206127_

Round 1

Reviewer 1 Report

Given the heterogeneity of available data on the antiproliferative activity of high dose (HD) SSAs, the authors undertook systematic review and meta-analysis to assess the current literature regarding the efficacy of increasing octreotide or lanreotide dose in patients progressive GEP NET after standard dose.  This is an important work for clinic application and well-written. I think this paper can be accepted after the following comments are addressed.

1.  What is standard dose?

2. Long-acting SSAs octreotide and lanreotide are widely considered effective and well tolerated first-line treatment for G1-G2 GEP NETs expressing sstr. Please compare its advantage over other approaches in the literature.

Author Response

Thank you for this review.

  1. What is standard dose?

Standard dose for octreotide long-acting release (30 mg every 28 days) and lanreotide autogel (120 mg every 28 days) is reported in lines 65-67.

  1. Long-acting SSAs octreotide and lanreotide are widely considered effective and well tolerated first-line treatment for G1-G2 GEP NETs expressing sstr. Please compare its advantage over other approaches in the literature.

Thank you for this suggestion. SSA have several advantages over other approaches in the management of NET including, convenient schedule, safe toxicity profile, excellent tolerability by patients, and syndrome control. These has been added to the manuscript in lines 60-65.

Reviewer 2 Report

The publication sent for review presents a review of publications on the treatment of neuroendocrine neoplasms with somatostatin analogues in an increased double dose. The works were analyzed, where the use of octreotide at a dose of 60 mg and lanreotide at a dose of 240 mg was described. In conclusion the HD-SSA is not the prefered treatment choice in patients with progression on standard-dose SSA. This is markedly more evident in studies carried out in the USA. However, a subset of patients with advanced age, long PFS, low Ki67/grading could benefit from HA-SSA.

The comments and observations described are in a way controversial in relation to the results of the CLARINET FORTE research.For this reason, the results of the CLARINET FORTE study should be cited in more detail in the discussion, which, nevertheless, presents great benefits for patients using a high dose of somatostatin analogues.

The work is of a valuable nature and after placing the above-mentioned minor note, it can be approved for printing

Author Response

Thank you for this suggestion. The results of the CLARINET FORTE study have been reported in lines 272-280.

Reviewer 3 Report

Dear Editor of Journal of Clinical Medicine

Many thanks for giving us the opportunity to review this article.

The authors described in this paper the activity of high dose of somatostatin analogs (HD-SSA) as antiproliferative agent for treatment Gastro-Entero-Pancreatic Neuroendocrine Tumors. They  collected data for both incidence density ratio (IDR) and the disease control rate (DCR).   Eleven studies counting 783 patients were included. The IDR was 62 new progressions of 100 patients treated with HD-SSA every one-years, and was significantly higher in USA centers and studies with more than 50 patients. The IDR was lower when a high rate of metastatic patients was present in the studies. They concluded that, the use of HD-SSA therapeutic approach is advisable in selected cases when other antiproliferative treatments are not feasible.

The author should carefully revised again the English language  throughout the manuscript, Example  in abstract t the treatemtns in the line 33 should be corrected.

The authors have already published this titled study in Digestive and Liver disease Vol.54, supplement 2, S134, May 01, 2022.  As poster, So they should indicated that.

The authors should report in this work as well:

·         Binding affinities receptor of somatostatin, octreotide and lanreotide.

·         Cell proliferation mediation receptor.

·         Collect of  some non-randomized clinical trials that evaluate the somatostatin analogs antiproliferative activity.

Best regards

Author Response

Thank you for this review.

  • The author should carefully revised again the English language  throughout the manuscript, Example  in abstract t the treatemtns in the line 33 should be corrected.
    • Thank you for this suggestion, English has been revised throughout the manuscript.

The authors should report in this work as well:

  • Binding affinities receptor of somatostatin, octreotide and lanreotide.

Thank you for this suggestion. Binding affinities of somatostatin, octretide and lanreotide have been reported in lines 72-74.

  • Cell proliferation mediation receptor. 

Thank you for this suggestion. The regulation of tumor cell proliferation mediated by sstr has been briefly reported in line 60-65.

  • Collect of  some non-randomized clinical trials that evaluate the somatostatin analogs antiproliferative activity.

Thank you for this suggestion. A commentary of the most relevant non-randomized trials about the antiproliferative activity of HD-SSA is reported in lines 268-289. A deeper review of the antiproliferative activity of standard dose SSA is felt to fall beyond the scope of the discussion of the present meta-analysis.

Round 2

Reviewer 3 Report

Dear Editor of  JCM

Many thanks again

The author has replied to all points.

best regards

Author Response

Thank you.

Best regards
